# USING KNOWLEDGE GRAPH IN ADAPTING LANGUAGE MODEL ON MATHEMATICAL TEXTS

**Olga M. Ataeva**
Federal Research Center 'Computer Science and Control' of the Russian Academy of Sciences
Moscow, 119333, Russia
`oli.ataeva@gmail.com`

**Natalia P. Tuchkova**
Federal Research Center 'Computer Science and Control' of the Russian Academy of Sciences
Moscow, 119333, Russia
`natalia_tuchkova@mail.ru`

## ABSTRACT

The subject of the study is the problem of adapting language models to scientific subject areas. The issues of expanding language models to mathematical subject areas are considered. It is proposed to use the knowledge graph of the subject area as a tool for 'tuning' the language model. To build the knowledge graph, the ontology of the subject area of the semantic library of mathematical subject areas and their applications LibMeta is used. Navigation through the subject area is carried out using the knowledge graph and is limited by the terminology of the thesaurus and ontology links. This approach allows using the knowledge graph to create a digital assistant in a recommender system, an agent for a language model, and to feed mathematical text data to a language model.

## 1 INTRODUCTION

Modern large language models (LLM) are being implemented in various information systems to provide a dialogue with the user in natural language. This radically changes the way information is extracted, accelerates the acquisition of necessary information and knowledge in certain cases. The influence of ChatGPT and LLM is enormous. Meanwhile, the university community is discussing the problems of devaluation of scientific work and plagiarism, in connection with the use of knowledge based on ChatGPT Ramil Malinka & all (2023) in the process of obtaining education. The study in this paper is devoted to the acquisition of knowledge in a specific mathematical subject area using LLM.

A well-known problem with using LLM is the difficulty of explaining and veracity of the conclusions, since the answer does not indicate the sources on the basis of which the answer is formed. As a rule, such sources cannot be specified even in principle, due to the complexity (closedness) of algorithms for processing large volumes of data. This is especially important when it comes to scientific knowledge, especially mathematical knowledge, which is needed in a wide range of classical and applied problems. In mathematical subject areas, it is important to rely on proven sources, to identify LLM hallucinations from true search results.

Some search engines that use LLM provide source references. The most professional and sophisticated product is *Perplexity AI* (https://www.perplexity.ai). *Perplexity AI* allows a question, including in Russian, and gives an answer in English and Russian with a link to Wikipedia and articles from *mathnet, matem.anrb, diffjournal.spbu* and other mathematical resources on the Internet. This version of the search engine, if we

do not talk about links to Wikipedia, certainly serves as an example of checking the authenticity (factuality) of the LLM answer. Naturally, it does not assess the correctness of the answer, from a mathematical point of view, but it does provide a link to the publication, which, as a rule, is the purpose of the search. The question of the evidence of mathematical conclusions Zong & Krishnamachari (2023) is not considered by us in this work.

However, despite the perfection of *Perplexity AI*, which works with the entire Internet space, there remains the problem of working with special requests in a specific subject area, in particular in Russian, where clarification is required, and not general information, which, in principle, the user can find himself, spending, of course, more time than with *Perplexity AI*.

As a rule, for scientific fields (and mathematics, of course), it is necessary to analyze a specific collection of articles, including archived ones, full texts of which are not in the public domain. These articles need to be collected in a digital library, processed, and only then can the result come to the attention of LLM. To extract knowledge from these texts, previously not found in the public domain, it is necessary to provide them for LLM training in a new subject area, i.e. to compile a corpus of articles and describe this set semantically. This endless process is still relevant, since new subject areas appear, and new interdisciplinary studies with new terminology are added to traditional ones, which means that language models need to be adapted.

In this paper, the problem of LLM adaptation is proposed to be solved by using the knowledge graph (KG) *MathSemanticLib* of the semantic library *LibMeta* Olga Ataeva & Tuchkova (2024b), bypassing which, LLM extracts the answer from the subject area of mathematics and its application. The result is achieved due to the fact that the KG represents structured data, relies on the ontology and thesaurus of the subject area.

The structure of the article is as follows: Introduction, Related works, Data model of the Kg MathSemanticLib, Supervised knowledge extraction example LLM answer and Conclusion.

## 2 RELATED WORKS

The issues of joint consideration of the problems of constructing KG and LLM arose naturally, as a continuation of the ideas of providing access to knowledge as structured data. The *Awesome-LLM-KG* (https://github.com/RManLuo/Awesome-LLM-KG) page presents a collection of links to papers and resources about unifying LLMs and KGs. It graphically displays the advantages and disadvantages of LLMs and KGs in the context of their mutual complementarity. The main idea is that the KG structure contributes to improving LLM reasoning, and the linguistic capabilities and generalizations of LLMs improve the understanding of the essence of knowledge in the KG. *Awesome-LLM-KG* also provides generalizations of research directions and applications of the results of unifying LLMs and KGs in searching, building dialog systems and AI assistants, and research methods. The synergetic nature of unifying LLMs and KGs is separately noted, which is based on the mutual enrichment of LLMs and KGs when they are combined.

In the work Shirui Pan (2024) a cyclic procedure of integration of the domain knowledge and LLM is considered, as a result of which the LLM response and the domain knowledge itself are corrected, which is closest to the idea of our research.

The authors Linhao Luo (2024) provide an overview of the weaknesses of LLM related to hallucinations. The authors see an improvement in the quality of inference in the use of KG for training LLM, but they note that this process is quite complex, since first a full KG must be constructed, and then LLM reasoning with graph constraints. The work Linhao Luo (2024) proposes a procedure for transforming KG for further traversal into LLM and for generating correct reasoning paths.

The option of creating a semantic description of mathematical concepts from school to university is considered in the work Samuel Debray (2025). Here Samuel Debray (2025) provides an overview of re-

search related to the attempt to reflect the process of cognition of mathematical subject areas and their reflection in digital resources. This idea itself has haunted the scientific community, starting with the GDML project Patrick Ion & Zheng (2019). The research Samuel Debray (2025) uses the *GloVe* (https://nlp.stanford.edu/projects/glove/) algorithm on a large corpus of mathematical texts to identify the frequency of use of terms and their relationships.

The comparison is made between Wikipedia terms in French and their English translations, and the use of words from the dictionary by participants (https://osf.io/dxg2w) with different mathematical backgrounds. In this way, the *GloVe* algorithms were tested and a relatively good correspondence between the *GloVe* vectors and human judgments was established. This study Samuel Debray (2025) is important for our discussions in terms of the participation of experts in assessing the results of the semantic representation of subject areas and the reflection of these representations in the processes of cognition of mathematical areas. Like the authors of Samuel Debray (2025), we use the opinion of experts, but when creating semantic images of subject areas, we rely on classical sources such as encyclopedias and monographs.

The research Ruiqing Ding (2023) is devoted to the description of subject areas. The data are given on how individual examples trained on corpora of specific subject areas achieve good results. However, it is noted that this is not enough to create a general approach for different subject areas. The authors propose the *KnowledgeDA* tool, a unified domain language model development service that can automatically generate a domain language model by performing three steps: (i) localize domain knowledge entities in texts using an embedding-similarity approach; (ii) generate enriched samples by extracting exchangeable pairs of domain entities from two representations of both the knowledge graph and the training data; (iii) select high-quality enriched samples for fine-tuning using confidence-based scoring.

A *KnowledgeDA* prototype for learning language models for two domains: healthcare and software development. This example of creating text corpora by subject area is quite problematic to extend to mathematical subject areas, since the original sources may differ radically in structure and presentation features (for example, the presence of formulas changes the process of text preprocessing).

The work Iz Beltagy (2019) is devoted to training the *SciBERT* (https://github.com/allenai/scibert/) model on scientific texts, where the possibilities of improving BERT after unsupervised pre-training on a large multi-domain corpus of scientific publications are demonstrated. The *BERT* model architecture Jacob Devlin (2019) is based on a multilayer bidirectional Transformer Ashish Vaswani & Polosukhin (2017) is used.

Our work presents a technology for constructing KG, starting from arrays of texts of scientific mathematical and interdisciplinary journals, to the integration of the obtained KG *MathSemanticLib* with LLM in the journal recommendation system in the environment of the semantic library *LibMeta* Olga Ataeva & Tuchkova (2022).

## 3    DATA MODEL OF THE KG *MathSemanticLib*

The approach used in this work is that first an ontology and thesaurus of the subject area are built, then a KG based on the ontology, and then LLM is used for communication in the library. The data structure and ontology model of the *LibMeta* library for the KG *MathSemanticLib* are described in the works Olga Ataeva & Tuchkova (2023a), Olga Ataeva & Tuchkova (2024a), here we will note only some of their properties, namely: integration of various sources (encyclopedias, monographs, journals, classifiers, thesauri, dictionaries, formulas) based on the ontology; construction of a KG based on the ontology; use of a KG for organizing a dialogue in the library.

Thesauri contain the main terms of LibMeta subject areas, linked by hierarchical and horizontal relationships. The data model in LibMeta is an ontology in OWL (which is represented as an RDF graph). Filling the library is a process of completing the ontology by integrating data in accordance with their descrip-

tions and metadata. The subject area is defined by forming a thematic subspace in the library ontology and establishing semantic links with the basic content of the library  Olga Ataeva & Tuchkova (2023b).

The mathematical encyclopedia [ME] [EM], the encyclopedia of mathematical physics [MathPh], the thesaurus of ordinary differential equations, the dictionary of special functions of mathematical physics and other Russian-language sources and components of the library [Ataeva3] are used as external basic taxonomies with which publications are linked. The creation and development of the LibMeta library [Ataeva0] is based on the integration of mathematical knowledge, both in the retrospective and prospective direction, by adding publications from various new subject areas of mathematics, related sciences and applications.

## 3.1   ONTOLOGY

The LibMeta digital library ontology defines the data structure.  The concepts that make up the LibMeta ontology are conventionally divided into concepts intended for:

– describing the content of a subject area;

– forming a thesaurus of any subject area;

– describing thematic collections;

– describing the task of integrating library content with source data from LOD.

Semantically significant connections are defined between these groups of concepts.  The following formal definitions are used to describe the ontology:

Definition 1. Library thesaurus $TH = \{T, R\}$, where T are terms and $R$ are the relationships between them.

Definition 2.  Library content $C = \{IR, A, IO\}$, where $IR$ are types of information resources, a set of attributes $A\{a_i\}$, information objects $\{IO\}$.

Definition 3.  Semantic labels $M = \{m_i\}$ of an information object are terms that are not included in the thesaurus, but are necessary for thematic division of information objects $IO$ within the subject area.

Definition 4. Semantically significant relationships of the library $P = \{P_i\}$ are the following main relationships:

$P_1(t, io)$ thesaurus term $\rightarrow$ information object;

$P_2(io, t)$ information object $\rightarrow$ thesaurus term;

$P_3(r, s)$ information resource $\rightarrow$ class of source objects, where information resource is a general definition for information objects stored in the system; thus, in fact, information objects are instances of information resources;

$P_4(a, sa)$ information resource attribute $\rightarrow$ property of source class;

$P_5(io, os)$ information object $\rightarrow$ instance of class from data source;

$P_6(m, io)$ semantic label $\rightarrow$ information object;

$P_7(io, m)$ information object $\rightarrow$ semantic label.

In fact, the concepts are divided into three categories: the first includes definitions of the concepts of the semantic library content, the second category refers to the definition of concepts necessary to support terms in the thesaurus of the subject area, and the third includes definitions necessary to define the processes of integrating the content of these resources. Based on these definitions, the main processes are described,

such as, for example, integrating data from different sources, categorization/classification, mapping different models of source data to a given subject area, constructing equivalence classes, etc. Fig. 1 shows the diagram of the article's links in the LibMeta ontology.

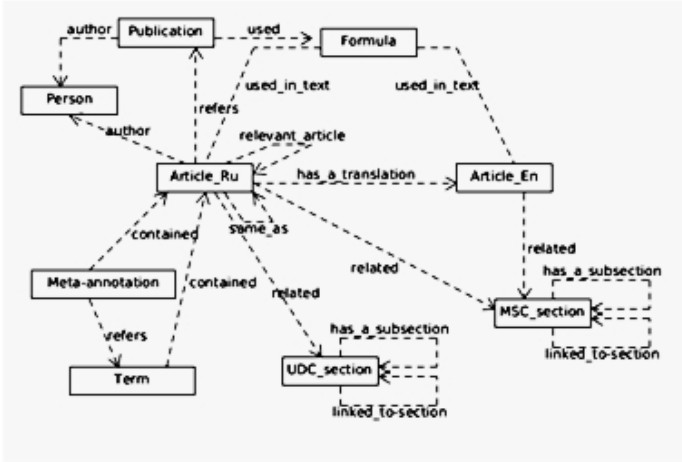

Figure 1: Diagram of LibMeta library data model.

## 3.2 COMPLETING THE ONTOLOGY AND INTEGRATING DATA

The task of adding new terms (new for this ontology) and links to the ontology arises during integration with new sources (publication arrays). These are, as a rule, terms from new tasks or applications in interdisciplinary research. Integration of new data into LibMeta is implemented by completing the ontology.

When placing publications in a semantic library, they must undergo preliminary processing, diagram Fig. 2.

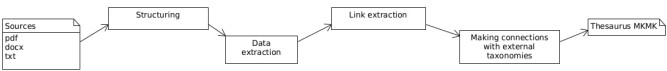

Figure 2: Stages of publication preprocessing.

Preprocessing options depend on the source data and may vary depending on the degree of structuring of the articles. Characteristic structures are known for mathematical articles, but it is necessary to identify the main terms and links Olga Ataeva & Tuchkova (2024a). At the *Link Extraction* stage (Fig. 2), semantically significant links of the library $P = \{P_i, i = 1, \ldots, 7\}$ are identified. If preprocessing has shown the presence of signs of belonging of the data to a certain subject area, then the publications are placed in the ontology and thesaurus of the subject area.

The task of adding new terms (new for this *LibMeta* ontology) and links to the ontology arises during integration with new sources (publication arrays). These are, as a rule, terms from new tasks or applications in interdisciplinary research. One of such typical examples is applications in equations of mathematical physics. Fig. 3 shows the scheme of adding terms from the journal MKMK Olga Ataeva & Tuchkova (2023b), thanks to which a new subject area, 'elasticity theory', was completed and integrated into the ontology.

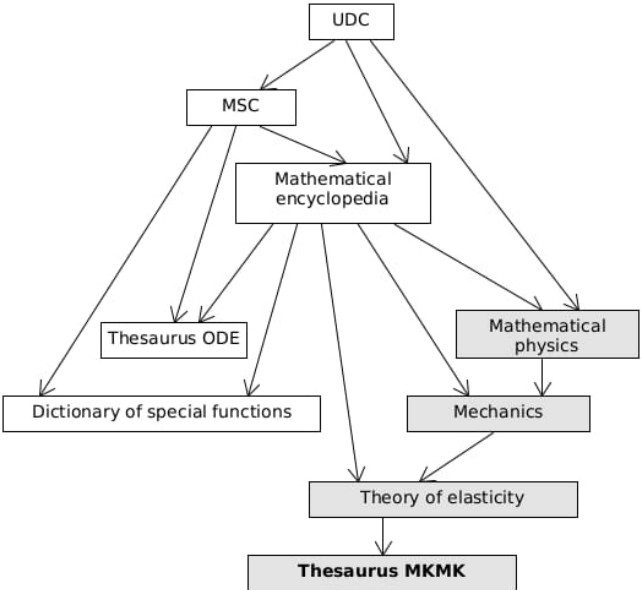

Figure 3: Connection diagram when adding a new subject area to the LibMeta ontology.

Completing the ontology, following the logic of the library data structure, affects the KG *MathSemanticLib*. Since the KG reflects the ontology connections, then when a new subject area appears in the KG of *MathSemanticLib*, a 'subgraph' appears.

### 3.3 KG *MathSemanticLib*

The *LibMeta* digital library ontology defines the structure of the library data. Each data element loaded into the library can be associated with an ontology node, which defines the position of the data element in the ontology. Based on the ontology links and the links defined at the design stage, a graph can be constructed. The subject area data can thus be represented as a KG, the structure of which is defined by the ontology, nodes (articles, terms, formulas) are instances of ontology elements, links are links of the subject area thesaurus. This is shown schematically as a three-level ontology in Fig. 4.

The construction of KG *MathSemanticLib* can be described in two global stages. At the first stage, a 'zero' version of KG is constructed from some source, and at the 'second' stage, the integration of the graph of incoming data with the general graph of the library occurs by establishing links with the thesaurus Olga Ataeva & Tuchkova (2024b). The 'zero' version of the graph KG *MathSemanticLib* is the KG of the mathematical encyclopedia Vinogradov (1977–1985), EM (2022), and the 'second' stage is the integration of an array of scientific articles. When completing the ontology, KG is also completed, that is, the 'second' stage is each subsequent stage.

The main stages of data processing for GC are closely related to the sources from which the data comes. Often the data is presented in an unstructured or semi-structured form. In our case, we consider, among other things, unstructured texts of Russian-language scientific articles. Nodes can be larger ontology objects Fig. 4, or objects - publication, term, person, formula.

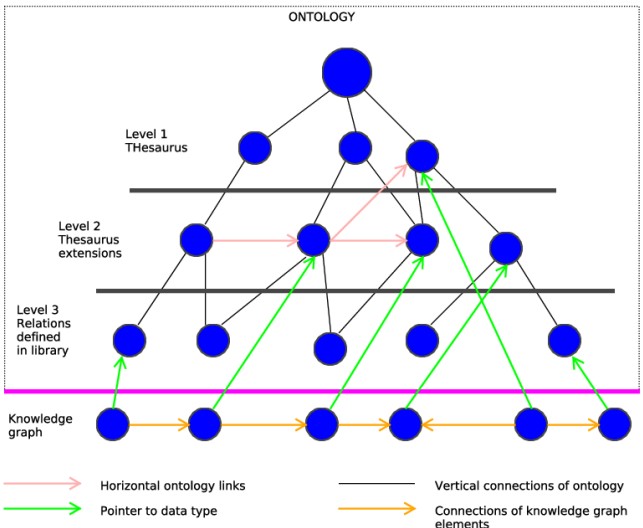

Figure 4: Scheme of three levels of the *LibMeta* library ontology.

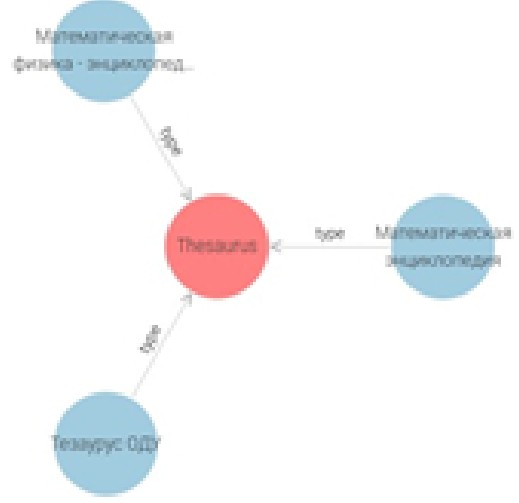

Figure 5: The central node of the MathSemanticLib thesaurus.

### 3.4 FORMULAS IN KG *MathSemanticLib*

The use of formula language in mathematical subject areas is a natural stage in modern dialog programs. One of the most authoritative databases of scientific publications zbMATH (https://zbmath.org) has long allowed

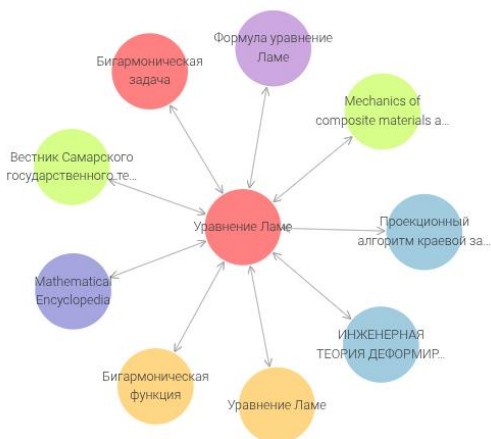

Figure 6: Fragment of the subgraph of the ME concept 'Lame Equation', KG *MathSemanticLib*.

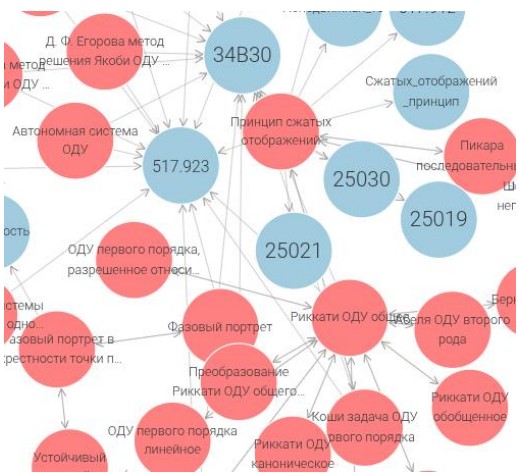

Figure 7: Subgraph fragment with MSC codes nodes, KG MathSemanticLib.

formula entry in the search line. Nevertheless, search by formulas remains one of the tasks in information technology, since it is important to consider the meaning of their use in a scientific publication.

The *LibMeta* library accumulates and integrates formulas from the mathematical encyclopedia Vinogradov (1977–1985), EM (2022), encyclopedia of mathematical physics Faddeev (1998), the thesaurus of ordinary differential equations Olga Ataeva & Tuchkova (2017), Eugenii Moiseev & Tuchkova (2005), dictionary of special functions and others into the ontology. The peculiarity of this integration is that the formula is saved with the context, and thus, a dictionary of formulas with links is formed, that is, the semantic image of the formula is saved. This approach allows you to include symbolic expressions in search queries. Since

the formula is a semantic object in *LibMeta*, it can also be a KG node, for example, Fig. 5, which shows the links of the formula with other KG objects of the *MathSemanticLib* library.

There are several types of work with formulas in the *LibMeta* library:

– selection of basic formulas related to concepts;

– selection of formulas related to the concept;

– selection of all formula links.

## 4 SUPERVISED KNOWLEDGE EXTRACTION EXAMPLE LLM ANSWER

Diving into the terminology of the subject area means accepting some non-standard terms established by specialists. For mathematical subject areas, it is important to reflect the meaning of formulas and terms when describing subject areas in digital libraries. Fig. 4–6 shows fragments of the *MathSemanticLib* KG for a concept, formula, classifiers.

Below is a comparison of responses from some language models.

### 4.1 EXAMPLES WITH CHATGPT, YANDEXGPT, LLAMA

*ChatGpt 4* and *YandexGPT 4 Pro* ( https://console.yandex.cloud/) coped with the query 'Lame equation', but gave only general information without links and formulas with an error, *Perplexity* and *LLAMA* (Large Language Model Meta AI https://app.hyperbolic.xyz/models/llama-3-3-70b-instruct) did not cope at all (Fig. 7-9).

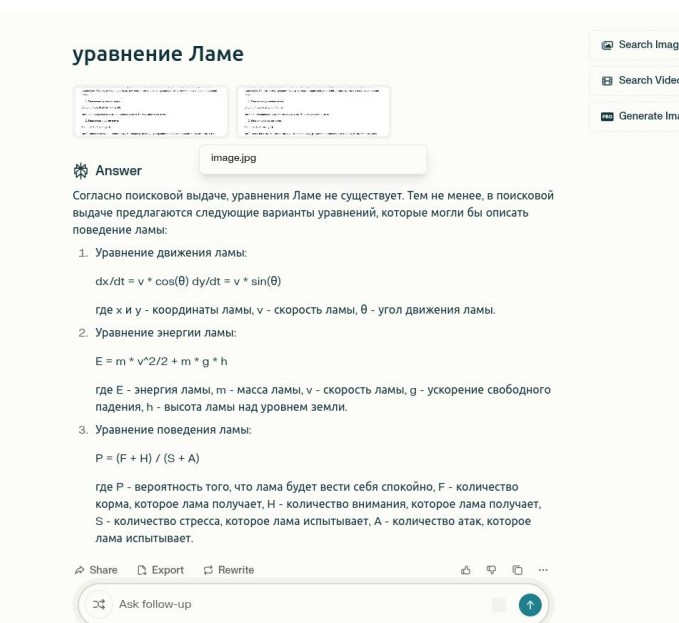

Figure 8: *Perplexity* answers to the query 'Lame equation'.

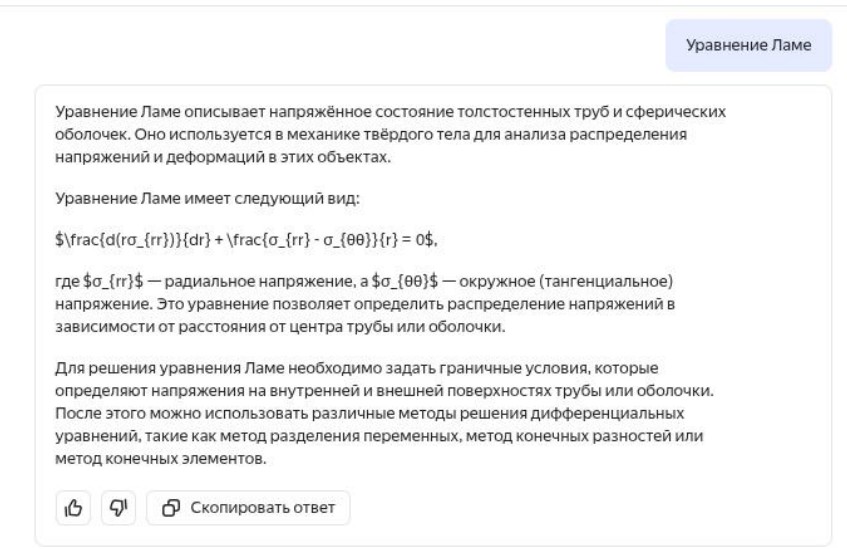

Figure 9: *Yandex* answers to the query 'Lame equation'.

The language model + knowledge graph allow using the generalizing capabilities of the language model to explore the domain graph in natural language. Questions formulated in natural language are transformed into *SPARQL* queries based on the ontological model of the graph. The model's answers are given based on the mathematical encyclopedia articles and their connections Fig. 10.

The model *Gemma2-9b-It* was used with following 3 questions. Gemma is a family of lightweight, state-of-the-art open models from Google, built from the same research and technology used to create the Gemini models. They are text-to-text, decoder-only large language models, available in English, with open weights for both pre-trained variants and instruction-tuned variants. Gemma models are well-suited for a variety of text generation tasks, including question answering, summarization, and reasoning. Their relatively small size makes it possible to deploy them in environments with limited resources such as a laptop, desktop or your own cloud infrastructure, democratizing access to state of the art AI models and helping foster innovation for everyone (https://huggingface.co/google/gemma-2-9b-it ).

**Question 1**: How many publications?

- Generated SPARQL:
  SE

LECT (COUNT(*) AS ?publications)

**Question 2**: How many concepts?

- SELECT (COUNT(DISTINCT ?concept) AS ?count)
  WHERE

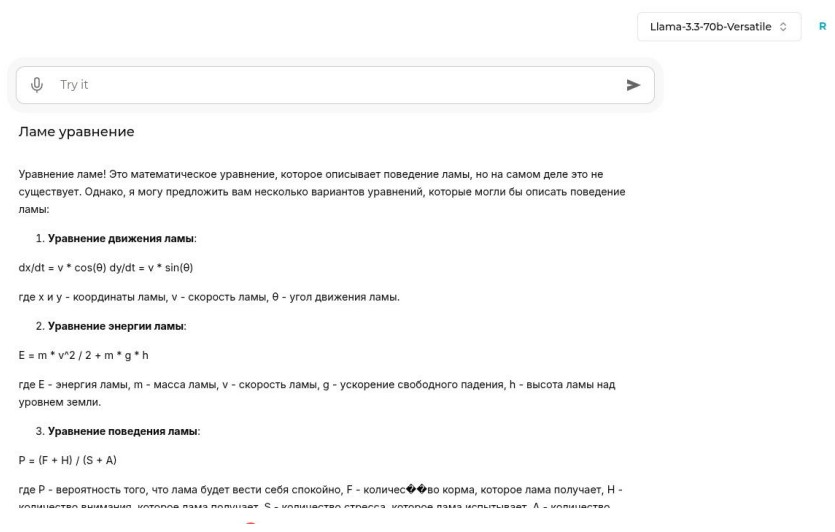

Figure 10: *LLAMA* Answers to the Query 'Lame Equation'.

> ?concept a :Concept .

**Question 3**: How many concepts are connected with Lame formula?

- SELECT (COUNT(DISTINCT ?concept) AS ?count)
  WHERE
  ?formula a :LameFormula .
  ?concept a :Concept .
  ?formula :connectedTo ?concept

## 4.2 STAGES OF GENERATING A SPARQL QUERY AND RESPONSE TO IT USING A GPT LIKE MODEL.

**Stage 1. Translation of text into SPARQL query.**

General format of the prompt for a model to generate a SPARQL query includes:

**Question (promt):** in natural language

**Scheme:** The graph data model is specified as an rdfs scheme, which is annotated with the rdfs:label and rdfs:comment properties, which contain a natural language description of the classes and properties of the ontology

**Instructions:**

- Generate a SPARQL SELECT query to query a graph database using the following ontology schema in Turtle format: {schema}

- Use only the classes and properties specified in the schema.
- Do not include any that are not explicitly provided.
- Ensure that all required prefixes are included.
- Output only the SPARQL query without any backticks or additional text.
- The question is: {prompt}

*Example 1.*

**Question (promt)**: in natural laguage  How many concepts are there?

**Scheme (fragment):**

....

http://libmeta.ru/thesaurus/concept/DE0002

http://www.w3.org/1999/02/22-rdf-syntax-ns#type

http://libmeta.ru/Concept .

http://libmeta.ru/thesaurus/ODU

http://www.w3.org/1999/02/22-rdf-syntax-ns#type

http://libmeta.ru/Thesaurus .

...

**Answer**

PREFIX rdfs: http://www.w3.org/2000/01/rdf-schema#

PREFIX rdf: http://www.w3.org/1999/02/22-rdf-syntax-ns#

SELECT (COUNT(?concept) AS ?conceptCount)

WHERE

?concept rdf:type ¡http://libmeta.ru/Concept¿ .

**Stage 2. Graph query**

Next comes the step of sending the generated query through the SPARQL access point and receiving the response. For the query specified in the example, the response comes in the form

'conceptCount': 'datatype': 'http://www.w3.org/2001/XMLSchema#integer', 'type': 'literal', 'value': '1221'

**Stage 3. Translate the answer into human readable format**

General format of the prompt to translate the response into humanreadable format includes:

**Question (promt): in natural language**

**Scheme:** The graph data model is specified as an rdfs scheme, which is annotated with the rdfs:label and rdfs:comment properties, which contain a natural language description of the classes and properties of the ontology

**Instructions:**

- Create a natural language response based solely on the results of a SPARQL query.

- You are an assistant who produces clear, human-friendly answers.
- Use only the provided information to build your response.
- This information is authoritative
- Do not question or alter it with your own knowledge.
- Ensure your answer reflects an AI assistants tone without adding any extra details.
- If no information is available, simply state that you don't know.
- Schema: {schema}
- Context: {context}
- Question: {prompt}

*Example 2.*

**Question (promt): in natural laguage  How many concepts are there?**

**Scheme (fragment):** ....

http://libmeta.ru/thesaurus/concept/DE0002

http://www.w3.org/1999/02/22-rdf-syntax-ns#type

http://libmeta.ru/Concept .

http://libmeta.ru/thesaurus/ODU

http://www.w3.org/1999/02/22-rdf-syntax-ns#type

http://libmeta.ru/Thesaurus.  **Context:**

{'conceptCount': {'datatype': 'http://www.w3.org/2001/XMLSchema#integer', 'type': 'literal', 'value': '1221'}}

**Answer**

There are 1,221 concepts in total.

**Stage 4. Correction of the query**

If an invalid request was generated in response to the first step, an attempt is made to correct it using the following instructions:

**Question (promt): in natural laguage**

**Scheme (fragment):**

....

http://libmeta.ru/thesaurus/concept/DE0002

http://www.w3.org/1999/02/22-rdf-syntax-ns#type

http://libmeta.ru/Concept .

http://libmeta.ru/thesaurus/ODU

http://www.w3.org/1999/02/22-rdf-syntax-ns#type

http://libmeta.ru/Thesaurus .  **Error:**

Error text

**SPARQL**

wrong query

**Instruction**

- The SPARQL query below
  {generated_sparql}
  is invalid.

- The error is: {error_message}

- Provide a corrected version of the SPARQL query without altering its logic.

- Include only the corrected query.

- No explanations or additional text.

- Do not enclose the query in backticks.

- The ontology schema in Turtle format is provided below: {schema}

*Example 3.*

**Question (promt): in natural laguage  How many concepts are there?**

**Scheme (fragment):**

....

http://libmeta.ru/thesaurus/concept/DE0002

http://www.w3.org/1999/02/22-rdf-syntax-ns#type

http://libmeta.ru/Concept .

http://libmeta.ru/thesaurus/ODU

http://www.w3.org/1999/02/22-rdf-syntax-ns#type

http://libmeta.ru/Thesaurus .

**Error:**

400: Error MALFORMED QUERY: QName 'lbm:ThesaurusConcept' uses an undefined prefix

**SPARQL**

PREFIX rdfs: http://www.w3.org/2000/01/rdf-schema#

PREFIX rdf: http://www.w3.org/1999/02/22-rdf-syntax-ns#

SELECT (COUNT(?concept) AS ?conceptCount)

WHERE

?concept rdf:type lbm:ThesaurusConcept .

**Answer**

PREFIX rdfs: http://www.w3.org/2000/01/rdf-schema#

PREFIX rdf: http://www.w3.org/1999/02/22-rdf-syntax-ns#

PREFIX lbm: http://libmeta.ru/

SELECT (COUNT(?concept) AS ?conceptCount)

WHERE

?concept rdf:type lbm:Concept .

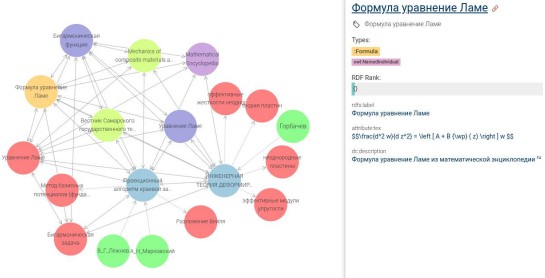

Figure 11: Graph for 'Lame equation', *MathSemanticLib* KG and answer.

Fig. 10 shows schematically T-T that the 'Lame equation' is selected and that it is linked to various objects of the library. These links arise because the formula in the *LibMeta* library is semantically linked through its context object.

## 5 CONCLUSION AND FINDINGS

The proposed approach to adapting LLM to a specific mathematical area of scientific Russian-language journals has been tested in the semantic digital library *LibMeta*. The experience of integrating LLM and the *MathSemanticLib* KG allows us to conclude that it is possible to organize a user dialogue with the library and create a digital assistant with the functions of a reader, author, editor, and reviewer of the journal.

It should be noted that when completing the *LibMeta* digital library ontology, it will be possible to ensure further integration of LLM and the *MathSemanticLib* KG, which means that the capabilities of a controlled Russian-language dialogue on mathematical topics will develop. At the same time, the ability to check the LLM output within the *LibMeta* content and external sources, which will continue to be integrated, will be preserved. Further research will continue in the direction of developing dialogue and recommender systems.

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
