# OpenReview forum: "Using knowledge graph in adapting language model on mathematical text"
_mathai.club/MathAI/2025/Conference — MathAI 2025 Oral_

### Official Review · Reviewer_rNXf · 2025-02-26
**The paper "Using knowledge graph in adapting language model on mathematical text" presents an interesting idea at the intersection of knowledge graphs (KGs) and language models for mathematical texts. The authors aim to leverage a knowledge graph to adapt or enhance a language model's understanding of mathematical content. While this is a promising direction – since mathematical text often contains complex notation and domain-specific knowledge – the execution in the paper falls short. Major concerns include significant methodological gaps and a lack of rigorous empirical validation. Key aspects of the approach (such as how the knowledge graph is constructed and integrated) are under-specified, making it difficult to assess the quality of the work or reproduce the results. Moreover, the evaluation is not convincing: the paper provides little in the way of quantitative benchmarks or comparative results to demonstrate the effectiveness of the proposed method. In summary, despite the potential of combining KGs with language models for math, the paper in its current form does not meet the standards of clarity and scientific validation expected for publication.**

**Rating:** 4
**Confidence:** 4

**Review:**

The quality of the work is hindered by missing methodological details and an inadequate evaluation. The idea of adapting a language model using a knowledge graph is sound in principle, but the paper leaves out critical details of the methodology. In particular, the authors do not clearly explain how they perform entity resolution – i.e., how mathematical entities or symbols in the text are identified and linked to the corresponding nodes in the knowledge graph. For example, if a formula or variable name appears in the text, it’s unclear how (or if) the system recognizes it as a specific entity in the KG, or how ambiguities in notation are resolved. Similarly, ontology alignment is not discussed: the knowledge graph likely has its own schema or ontology of mathematical concepts, but the paper doesn’t describe how this ontology was aligned with the language model’s internal representations or with any standard mathematical ontology. This raises the question of whether the language model truly understands the KG data or if it’s just being exposed to additional text without structured integration. The knowledge graph construction workflow is also glossed over – we don’t know what sources of data were used to build the KG, how triples (facts) were extracted or curated, and whether any verification or cleaning steps were done to ensure the KG’s quality. For instance, did the authors automatically extract definitions and theorems from a corpus of math documents to populate the KG, or did they use an existing ontology? The paper is silent on this workflow, leaving readers in the dark about how the KG was obtained and how reliable it is. Additionally, the preprocessing steps for mathematical texts are barely mentioned. Mathematical documents often include LaTeX formatting, special symbols, and structured elements (like equations, theorems, proofs) that a language model would need to handle carefully. The paper should detail how it preprocesses such input (e.g., converting LaTeX to a consistent format, isolating math notation from natural language, tokenizing equations, etc.), but it provides no such description. These omissions make it impossible to reproduce or verify the approach, and they cast doubt on whether the authors fully implemented the system they propose. Without clear explanations of these steps, the methodological soundness of the paper is in question.Furthermore, the empirical validation in the paper is very weak. There is a lack of quantitative benchmarks and a rigorous evaluation framework to support the claims. The authors do not report standard evaluation metrics or compare their approach against baseline models. For example, one would expect to see comparisons of the language model’s performance with and without the knowledge graph on some relevant tasks (such as mathematical question answering, theorem proving assistance, or text understanding tasks), using metrics like accuracy, F1, perplexity, or others relevant to the task. Instead, the paper’s evaluation is either absent or limited to a few examples without any statistical analysis. There is no evidence of the authors testing their adapted model on a benchmark dataset of mathematical problems or documents. They also don’t provide an ablation study to show how much the knowledge graph component contributes to any improvement – for instance, by turning off the KG and observing performance drop, or by varying the quality/size of the KG. The evaluation lacks rigor: no baseline comparisons (e.g., comparing to a standard language model fine-tuned on the same mathematical text without KG assistance), no error analysis, and no discussion of statistical significance or variability. As a result, the claims that the knowledge graph adaptation improves the language model are unsubstantiated. This is a serious shortcoming in quality: without solid empirical evidence, the work’s validity is doubtful. In sum, the paper does not meet the expected standard of experimental validation, making it hard to trust its conclusions.

The clarity of the paper suffers primarily because of the missing details described above. The writing itself is mostly readable in terms of grammar and structure, but key components of the approach are not explained clearly, which undermines the reader’s understanding. Important terms and processes are either mentioned only briefly or not at all. For instance, the concept of “using a knowledge graph to adapt a language model” is intriguing, but the paper doesn’t clearly articulate how this adaptation is performed (e.g., is the language model fine-tuned on text generated from the KG, or does the model query the KG at runtime to fetch relevant information?). Because such explanations are absent, readers are left to guess the intended methodology. This vagueness makes the contribution hard to comprehend. On a positive note, the general motivation of the work (why integrate a KG with an LM for math text) is stated and easy to grasp, and the problem addressed is clear. However, the solution description is too high-level. As it stands, the lack of transparency in crucial parts of the methodology makes the paper difficult to follow and understand in depth.

The paper’s originality is moderate. Adapting or fine-tuning language models for domain-specific text (like mathematics) is a well-known challenge, and using structured knowledge (in the form of knowledge graphs) to aid language understanding is also an active research area. The authors’ idea to combine these – a KG-powered adaptation for mathematical language modeling – is a logical next step, and to the best of my knowledge, applying this idea specifically to mathematical text could be somewhat novel. There have been knowledge-enhanced language models in other domains (for example, biomedical text or general commonsense reasoning, where a KG is used to provide additional context or constraints to a language model), but we haven’t seen much in the specific area of mathematical text processing. In that sense, the paper could claim a niche originality in targeting the math domain. However, the paper does not do a good job of positioning itself relative to prior work. It either does not cite or does not discuss related efforts in integrating KGs with language models. For instance, there are prior works on injecting knowledge graphs into transformer models, or on creating knowledge graphs from mathematical literature, which the authors should have compared or contrasted with their approach. Without this context, it’s hard to tell how much of the technique is truly new versus a re-application of existing methods. The contribution thus feels incremental: it’s essentially taking known concepts (KG + LM) and applying them to a new domain (math text) without demonstrable technical innovation. That said, if the authors had provided more details, we might discover novel aspects (perhaps a new algorithm for linking math text to KG, or a new way to integrate KG triples into the language model’s training data). But since those details are lacking, the perceived originality remains limited. In summary, the work has an idea that is potentially original in context but fails to highlight unique insights or improvements clearly. The novelty is modest and is overshadowed by the execution issues.

The significance of this work is also limited in its current form. The problem area – improving language models for mathematical text – is certainly significant in principle. Mathematical texts (like research papers, textbooks, or problem descriptions) are difficult for generic language models to parse because of specialized terminology, symbols, and structured reasoning. A successful method to adapt language models using a knowledge graph of mathematical knowledge could greatly benefit tasks such as automated theorem proving, math tutoring systems, or semantic search in mathematical literature. However, due to the weak evaluation and unclear methodology, the paper does not convincingly demonstrate a contribution that advances the state of the art. If the approach had been validated with strong results (e.g., showing clear improvements over baselines on a math QA dataset or significantly better comprehension of formulas), then it could be a notable step forward. Unfortunately, with no solid evidence presented, it’s unclear if the approach actually works in practice or yields any improvement at all. This severely diminishes the work’s impact. Additionally, because the methodology isn’t reproducible as described, other researchers or practitioners cannot easily build upon this work – this further limits its significance to the field. In its current state, the paper is more of a conceptual proposal than a proven contribution. As a proposal, it highlights an interesting direction (which has inherent importance), but as a research contribution, it lacks the substance needed to influence future research or applications. In conclusion, while the topic and intended application are important, the paper’s actual contribution seems marginal. It neither offers a clear scientific insight nor a validated tool that others could use, thus the significance is minimal until the authors strengthen the work.

---

### Official Review · Reviewer_7eCJ · 2025-02-27
**No scientific result is present in the paper submitted**

**Rating:** 3
**Confidence:** 4

**Review:**

The paper claims a method of using a knowledge graph in adapting language model on mathematical texts. The claim is largely unsupported. No method of adapting a language model on mathematical texts is presented in the paper. Instead, the paper describes the specific ontology and a knowledge graph and provides some answers generated by a dozen of LLMs to a specific question. In one of answers, a LLM uses a knowledge graph. No description is provided as to how specifically a LLM is integrated with a knowledge graph. No statistical data on the results are provided.

Additionally, the paper is highly sloppily written. The paper contains multiple unsupported marketese claims like “The most professional and sophisticated product is Perplexity AI (https://www.perplexity.ai).” (line 36-37) or “the perfection of Perplexity AI, which works with the entire Internet space,” (line 45). Citations are highly sloppy and unreadable (a citation should specify the author’s last name, not letters with dots like “M. & B. (2023)”), the authors of the paper should fix their .bib file and try to somehow read the paper they have written.

---

### Official Review · Reviewer_vk5V · 2025-02-27
**There are problems in the "Using knowledge graph in adapting language model on mathematical text" paper**

**Rating:** 6
**Confidence:** 3

**Review:**

This paper is devoted to solution of such important task as combining knowledge graph and large language models for reasoning about mathematical texts. Development of digital assistant prototype allows authors to solve this task.

This paper has the following disadvantages:
1) Authors should clearly describe role of Gemma2-9b-It model in translation questions from natural language to SPARQL queries. Also authors can mention similar application of large language model described in the "Ontology engineering with Large Language Models" paper (DOI: https://doi.org/10.1109/SYNASC61333.2023.00038 ). Moreover authors can add scheme of proposed digital assistant as software system to clarify its application.
2) Presence of grammatical errors in the paper, for example, use of "humanreadable" word instead of "human-readable" word at lines 541 and 542, use of "laguage" word instead of "language" word at lines 518, 561, 577 and 603.

---

### Decision · Program_Chairs · 2025-03-08

**Decision:**

Accept (Oral)

**Comment:**

Your article has been accepted and you can give a talk on the article. All articles will be sorted by rating and within the available conference places one author from each article will be invited. If there are not enough places, then you will either have the opportunity to speak remotely or come at your own expense!